# 'The community lives on sleeping medication and antidepressant[s]….': Health care workers' experiences of mental health service provision in rural South Africa

Divan Rall ‡*, Leslie Swartz&

Department of Psychology, Stellenbosch University, Private Bag X4, Matieland, Stellenbosch, Western Cape, South Africa

& This author contributed with supervision, technical attention and modifications, providing input to the literature review, discussion and conclusion.
‡ This author conceptualized and drafted the manuscript.
* divanrallphd@gmail.com

## Abstract

South Africa's public health care sector has changed dramatically, with a shift away from focusing curative health care primarily at tertiary health facilities towards the expansion of services into primary and secondary care health levels. This integration process has presented with a multitude of serious contextual and systematic difficulties. Limited studies have explored the integration of mental health services in poorer parts of the country where resources are scarce. We studied access to, and provision of mental health care in a predominantly rural area of the Eastern Cape to determine how primary and secondary care facilities that employ primarily generalist practitioners deal with psychiatrically ill patients and those facing life challenges with mental health implications. Data were collected through a descriptive data sheet and once-off semi-structured interviews. Narratives were imported into Atlas.ti 22 and analyzed through Thematic Analysis. Our results suggest that health care workers at primary and secondary health settings experience a high workload with diversity in patient population, and challenges at facility level, utilizing various means to deal with patients. Lastly, our participants suggested helpful strategies and proposed changes for more effective treatment. The integration of mental health services at primary health care and secondary hospital level introduces challenges to access and provision capacities. Despite facing intense challenges, at times our participants have found ways to adapt and work around the challenges. The decentralization and integration of mental health care into lower care sectors demonstrates a situation of great stress and, often, neglect. The mismatch between what is theoretically available and what is actually provided is worrying, and all mental health advocates and specialists should be engaged in advocacy about this. Ideologies of inclusion ring hollow when inclusion does not in fact occur.

**Data availability statement:** All data underlying the findings described in the manuscript are within the manuscript itself.

**Funding:** The author(s) received no specific funding for this work.

**Competing interests:** The first author was the PI and clinical psychologist in the DBNLM public health sector at the of conducting the research. This dual role may have influenced the manner in which the information was collected, analyzed, and interpreted. We wish to further declare that the Eastern Cape Department of Health (ECDoH) granted permission to conduct the presented research on the condition that ECDoH is presented with the work before publication.

## Introduction

Mental illness is a major global health issue, and mental health systems worldwide are marked by serious gaps in governance, resources, services, information and technologies for mental health. Mental health care is negatively affected by various factors like limited access to adequate care, low levels of MH knowledge, and stigma. These issues may be exacerbated in low- and middle-income counties (LMICs) where lacking services, limited skills, and insufficient funding for MH are prominent and significantly below what is needed. Yet, LMICs, such as South Africa, mark the highest rates of mental illness and the majority of the global population [1]. In attempt to reduce the gap in MH care South Africa's public health care sector has changed dramatically over the past 30 years, with a shift away from focusing curative MH health care primarily at tertiary health facilities towards the expansion of services and integration of holistic MH care at primary and secondary health levels. This is in keeping with global trends and recommendations [1–3].

Local literature suggests that the lifetime prevalence of a South African living with a diagnosable mental disorder is over 30%, or approximately one out of three citizens [4]. While as many as 80% of South Africa's MH care service users rely on care within the public health system [5], it is only staffed by approximately 20% of the country's medical workforce [6]. In principle, the integration of MH care into primary health care (PHC) and secondary health settings should allow for a more comprehensive and accessible health care to service users by adequately trained health care workers. The National Department of Health, for example, developed a National Mental Health Policy Framework and Strategic Plan (NMHPFSP) 2013–2020 [7] to guide the country in areas of MH promotion, the prevention of mental illness, treatment and rehabilitation interventions to improve MH services locally. The NMHPFSP 2013–2020 elaborately describes goals that were set for MH services in the country by 2020. It stipulates, among other aspects, that by 2015 all health staff working in general health settings would have received basic MH training, and ongoing routine supervision and mentoring would be provided. Furthermore, trained non-specialist health workers would deliver evidence-based psychosocial interventions, with supervision and support from specialists – a concept known as task-shifting. It was envisaged that by 2020, services such as specified MH interventions would have been rolled out and facilitated through task-shifting model, that MH training programmes will have been established for generalist health worker at lower levels of care, and that supervision and support functions will have been executed by specialist workers for MH staff at lower level facilities [7].

The reality is that the integration process has presented with a multitude of serious contextual and systematic difficulties [2,8]. At present, the public health sector is experiencing significant strain due to serious resource constraints [9,10]. Furthermore, in practice, generalist health care practitioners may not be specifically trained to deal with an array of MH cases, or may lack particular interest in doing so. In addition, first line clinicians at lower level care facilities have multiple endemic conditions (e.g., poverty-related conditions, such as malnutrition and infectious diseases, as well as HIV/AIDS) to

contend with [5,11,12], and an increase in MH care load comes as a possible additional work burden with unintended consequences [13]. In the context of higher workload and inherent limited human ability to deal with an excess of work, generalist health workers may experience difficulties when confronted with psychiatric patients [13–16], with consequent treatment gaps.

A body of literature has investigated approaches to addressing the MH treatment gap in the South African context [5,8,9,12,17]. In line with recommendations put forward by the World Health Organization (WHO), attempts have been made to divide MH care responsibilities between generalist and specialist clinicians to improve access to, and provision of services locally [18]. This task-shifting approach provides the hope of narrowing the gap in MH care [5,9,18–20]. Le et al.'s systematic review [21] of interventions in 18 countries, mainly in Africa and South Asia demonstrate that, especially in LMICs, the redistribution of MH care from specialist to generalist level and the implementation of evidence based interventions can be a very complex matter. Though it may be possible to train frontline providers in task-shifting, there are large implementation barriers (e.g., patient and broader societal factors, and administering quality control measures) that are challenging to modulate or intervene upon [2,22,23]. Furthermore, attempts have been made to up-skill generalist workers who are now required to assist the health system, and one another, to better treat and manage MH service users at lower level facilities. Local literature has suggested some favorable outcomes in equipping generalist workers with the basics to deal with patients that present with mental illnesses [9,24,25]. In such instances, however, generalist workers may require close supervision and ongoing education to provide MH care [9,3,26], and this process may be time-consuming and not universally feasible across the entire country, given the maldistribution of resources.

Going back more than twenty years, a number of studies have documented challenges in integrating MH care into PHC in South Africa, including in the Western Cape province, which is one of the wealthier provinces, commonly considered to have the best functioning health system [27]. More recently, also in that province, van Heerden et al. [28] noted difficulties with integration. They found that lower level care facilities are commonly poorly equipped, understaffed, and unwilling to take on the extra burden of MH responsibilities, which can lead to a diminished health service. Directives to more comprehensively integrate MH care throughout all sectors of treatment delivery appear to create pervasive cross-dimensional issues across systems. It is inevitable that gaps will result, that may ultimately impact negatively on service provision in facilities, and thus on health workers and service users.

Limited studies have explored and reported on the integration of MH services in poorer parts of the country, such as the Eastern Cape province, where resources are scarce [29–31]. The Eastern Cape province had 49 registered psychiatrists, and 348 psychologists (clinical, counselling, and educational) [32] to service a population of over 7.2 million in 2022 [33], whereas the Western Cape had 265 registered psychiatrists and 1204 psychologists in similar categories of registration [32] to service 7.4 million people [34]. Furthermore, there are serious concerns with quality and governance regarding service provision in the Eastern Cape [35]. Most service users in this province are likely seen and treated by generalist health care workers who may have limited knowledge of MH and may be working in strained health systems, with seriously limited specialist support, and potentially experiencing major challenges meeting job-related demands in general [13].

We conducted research on access to, and provision of MH care in a predominantly rural area of the Eastern Cape. We were interested in knowing how primary and secondary care facilities that commonly employ generalist practitioners (e.g., nurses, generalist doctors, pharmacists, and other allied health workers) with limited, on site, MH clinicians, deal with psychiatrically ill patients and those facing life challenges with MH implications. As part of a larger study [13,36] we present detailed qualitative data on interviews with frontline personnel.

## Research methods

### Ethics statement

The study complied with the principles expressed in the Declaration of Helsinki. The study was approved by the Health Research Ethics Committee of Stellenbosch University [Reference number: S21/07/117 (PhD)]. Access to study public sector workforce and service users was received from the Eastern Cape Department of Health (ECDoH). Participants

provided written informed consent to participation in the study and having their experiences published in academic literature. Participation was voluntary and participants could withdraw from the study at any stage of the process without any consequences.

## Study setting

We conducted the project in the public health care sector in the Dr Beyers Naude Local Municipality (DBNLM) area of the Eastern Cape province, South Africa. The Eastern Cape is the third most populous province in the country and one of the poorest. The DBNLM is one of seven municipal regions that make up community and service sectors in the Eastern Cape. The DBNLM area primarily consists of rural towns, farms, and smaller settlements. DBNLM had a population of just over 82 000 people in 2016 [37]. Data were collected from five PHC clinics, one community day care (CDC) facility, and three secondary care hospitals in the area. The first author is familiar with the area, as he worked as a clinical psychologist in the public and private sectors at the time of the research.

## Study design

The larger project investigated aspects of access to, and provision of public MH services by utilizing both quantitative and qualitative research techniques. We utilized an explorative qualitative research design for the currently presented part of the study. The presented work is the first to exclusively report qualitatively on the experiences of a group of frontline health workers that provide MH care in the area. We deemed the qualitative exploratory approach particularly suited for our project as it allowed us to investigate a phenomenon in an area that has not yet been thoroughly studied. This research design would potentially inform us of the subjective experiences encountered by participants, which shape the way that MH care is accessed and provided in the DBNLM community.

## Reflection on the positionality of the authors

Both authors are qualified clinical psychologists. The first author worked as the only clinical psychologist for ECDoH in the area at the time of conducting the research and developing the presented paper. This placed the first author in a dual role as clinician and researcher. It cannot be ignored that this duality and the first author's personal views may have impacted the formulation of this paper and the effects on his clinical duties. In order to mitigate such effects, the first author was routinely supervised by the second author who was not an insider. Furthermore, the presented work was informed by and critically evaluated against existing local and international literature in the field. The first author being an insider, may also have had a beneficial impact on the paper; as he was able to formulate and write the presented paper with firsthand knowledge of a rural MH care phenomenon in a setting that has received almost no attention in professional literature. It allowed the first author to be a voice for his colleagues, who more recently, are working in "tough times" [38] in public health care facilities in South Africa and often struggle in silence, and now chose to participate in the study to express their experiences that seemingly differ in many ways as we see from studies elsewhere in the world and our country [39,40].

## Participants and data collection

Participants in this part of the study were public health care workers who provided some form of care to MH care users at the facility where they were employed (please see Table 1 for participant population description). Participants were recruited by means of purposive sampling and approached via advertisement posters and brochures at facilities between January and March 2022. Participants who were interested in joining the study completed a brief identity and contact note, whereafter they were telephonically contacted by the first author, who was the primary investigator (PI) of the project.

Data were collected through a brief descriptive data sheet and an in-person once-off semi-structured interview of approximately 60 minutes with the first author (please see S1 Text for the descriptive data sheet and S2 Text for the

Table 1. Summary of participant descriptive information.

| Participant | Age | Position | MH training prior to employment at current facility* | MH refresher course (past 2 years)* | Years' experience |
|---|---|---|---|---|---|
| Lily | 49 | Social worker | No | No | 25 |
| Cottonwool | 28 | Medical officer | Yes | Yes | 4 |
| Participant X (This participant chose a pseudonym, "Valium" that is the name of a prescription psychotropic drug. Peer review identified that the originally chosen pseudonym may confuse the reader in the context of the presented work and requested that the pseudonym be changed. The pseudonym was changed to Participant X.) | 29 | Pharmacist | Yes | No | 5 |
| Diena | 56 | Professional nurse | No | No | 10 |
| Fern | 57 | Enrolled nurse | No | No | 33 |
| Sterretjie | 54 | Nurse | No | No | 35 |
| Niesie | 35 | Professional nurse | Yes | No | 10 |
| Anna | 45 | Pharmacist | Yes | No | >20 |
| Flobiza | 47 | Professional nurse | No | No | 6 |
| Littlerose | 41 | Professional nurse | Yes | No | 18 |
| Lungi | 52 | Professional nurse | No | No | 11 |
| Mrs Jones | 33 | Enrolled nurse | No | No | 7 |
| Tikos | 58 | Medical officer | Yes | Yes | >20 |
| Nadine | 38 | Professional nurse | Yes | No | 10 |

*This information forms a part of the data reported in Rall & Swartz [42]. We discuss more of this below; the vast minority of participants felt that their exposure to MH care training was sufficient and the majority of participants reported that there were areas of their ability to provide care to MH care user that need further training or development.

interview guide). Interviews occurred between 2022 March 11 and 2022 July 08. Interviews were scheduled at a mutually convenient time at the participant's office/place of work. Afrikaans (76.6%), isiXhosa (21.3%), and English (2.1%) are the prominent languages spoken in the area [41]. All participants chose to conduct their interviews in either Afrikaans or English, both languages in which the first author is fluent. Interviews were audio-recorded, and transcribed verbatim. Participants chose their own pseudonyms to protect their identity and enhance anonymity. Interviews were conducted with 14 health workers; by this stage we believed that data saturation had been reached.

## Data analysis

To enhance analytical rigor, the research followed systematic processes to approach participants and collect the data as described above. The first author then imported the narratives that were transcribed into ATLAS.ti version 22 to orientate and structure the data for analysis. The first author had firsthand knowledge of the interview content, as he was the PI of the study. The first author further immersed himself with the data by replaying audio-recordings and reading the transcribed audio-recordings. The first author used thematic analysis [43] as the primary approach to analyze the data. The first author analyzed the data and derived the codes used to build the themes presented. The codes derived through organizing the data served to group, but also to confirm different participants' experiences with another. The authors regularly met for supervision meetings for the PI to reflect on, discuss, interpret, and collaboratively orientate the data. Supervision meetings served as debriefing and a space for critical thinking about the research to facilitate credible reporting of

participants' experiences. Furthermore, during analysis we found that participants frequently presented extended narratives or interactions during interviews. We judged that the longer phrases and interactions between participants and the PI provided the foundations for participants to express key issues, namely, 1) their experiences regarding issues of MH services, but also to display, 2) the nuanced complexities encountered within the systems where they work [44] (citing Creswell, 2012). Yin (2011) cited [44], concurs that lengthier extracts can better represent participants' experiences as they contain larger bodies of data. Our findings, therefore include elaborate narratives that aim to illustrate participants' experiences of the intertwined and often linked complexities involved when providing MH services in the DBNLM region. The elaborate narratives presented in the results also serve to contextualize participants' experiences to reduce superficiality that may lead to diminished credibility.

## Results

The interviews with the participants, and subsequent data analysis, produced themes and sub-themes that describe some of the experiences that health care workers encounter when providing MH care in the area (please see Table 2 for a summarized view of the themes and sub-themes that emerged after data analysis).

### Theme 1: Mental health services in a context of high workload

Many of our participants experienced facing a significant workload that impacts services. Niesie suggested that this may be due to the integration of MH care into other services:

> Niesie: *At first mental health care used to be separated from the other programs, but now it has been integrated into the system. For me personally, I feel that is where the, the missing of, of, of patients, because now there's not one person responsible. It's everybody and not everybody is really fond or interested in mental health. So uhm, they'll – they don't go the extra mile for mental health patients, yeah.*

> PI: *So once the integration of mental health into general care occurred – …*

> Niesie: *Lots of, lots follow ups now occurred as a result of the integration because remember with integration, it means that the patient will enter into the facility and be stationed between the other patients or so. So the, the waiting period*

**Table 2. Themes that emerged from data analysis.**

| |
|---|
| **Theme 1: Mental health services in a context of high workload** |
| **Theme 2: Diversity of mental illness presentations** |
| **Theme 3: The challenges experienced at health care facility level** |
| • Sub-theme 1: Health system challenges affecting mental health services |
| • Sub-theme 2: Lack of resources |
| • Sub-theme 3: Limited knowledge for dealing with complex cases |
| • Sub-theme 4: Patients who present a physical danger to others |
| **Theme 4: Means of dealing with and managing mental health care patients** |
| • Sub-theme 1: Pharmacotherapy and relying on sedation |
| • Sub-theme 2: Long distance consultation and care delivery |
| • Sub-theme 3: Desperate improvising |
| **Theme 5: Helpful strategies and proposed changes for more effective treatment** |
| • Sub-theme 1: Ongoing in-service training on mental health |
| • Sub-theme 2: Psychiatrically trained staff post matching and multidisciplinary team approach |
| • Sub-theme 3: Community-based education and outreaches |

*for these patients is not so fast as previous years. So now they have to wait to be seen and because of that uhm, most of them uhm, have short temper, so then they don't come to the clinic now. They don't want to wait that long. So we miss a lot of people.*

Littlerose similarly noted that they are required to treat many patients within a limited time:

*The numbers are just so many, so big. You can't always take your time with a patient and you know there is perhaps something deeper that is bothering that patient and that has an impact on his health, but you can't always sit with the patient for half an hour and support him a bit and all those little things like that because you have to – there are twenty others out there waiting for you oh, my, you're taking so long now and, oh it's going slowly.* (Littlerose) [Translated from Afrikaans to English]

### Theme 2: Diversity of mental illness presentations

The participants reported different types of mental disorders presenting to lower level care facilities in the area. The narratives suggest that health care workers are required to care for patients across the age spectrum who experience an array of difficulties, including, social and relational problems, financial stressors, behavioral disturbances, mood disorders, substance abuse and psychosis, and trauma.

Lily described that, as a social worker, they treat and manage social problems that affect the behavior and mental health of patients and their families:

*Our focus is mostly on support and on social problems. OK! Social problems, which in the end lead to domestic violence, and that type of thing. So the things that we deal with the most, that are referred to the most are your parasuicides, people with depression, neglected children and the elderly, which we then eventually try to place.* (Lily) [Translated]

Participant X explained how an increase in youth MH care user cases present and require admission at the facility where they work. Participant X elaborated on how these patients 'have lost their minds' and can be a physical or sexual risk towards other patients and staff during admission:

*More and more younger children – fourteen, fifteen, sixteen-year-old kids who come in with, then they are completely out of their minds then all of a sudden they have to use medication for it. That's most of the stuff I've seen here. Those patients obviously have to lie in the wards with the other patients. Those patients assault other people in the wards, assault nurses or assault them sexually or verbally or whatever.* (Participant X) [Translated]

Niesie reported that they also encounter persons who become suicidal because they are depressed: *'So people are still committing suicide... when they feel very depressed or low....'*

Fern explained that they also care for patients with bipolar mood disorders who require pharmacological treatment from the doctor: *'Bipolars and so on, those are also the types of patients we deal with, so the doctor they prescribe the medication.'* [Translated]

Substance use and substance-induced psychosis seems to be a prominent presentation at facilities that health care workers are required to treat. Cottonwool explained how drugs are a major problem in the community: *'They're doing drugs. They're becoming psychotic. They are a burden on this community. Substance – eighty to ninety percent of our mental health care users in this community is substance induced. Big problem.'*

Trauma cases also present for treatment at facilities where health workers, such as Tikos, are responsible to facilitate care. Tikos reported:

*I had a patient who was – her boyfriend was shot. Here, was shot and she was there. She was there. She, she, she watched that happen. So she came to me…. She mentioned to me that at night these – the incident comes back and back and doesn't make her sleep properly.* (Tikos)

**Theme 3: The challenges experienced at health care facility level**

**Sub-theme 1: Health system challenges affecting mental health services.** Participants experienced systematic health system challenges that affected the MH care services they could provide.

Fern described occasions when the medication depot that distributes pharmacological treatments to their facility would be required to do a stock take. During this time medication delivery is diminished – facilities may run out of stock, or need to borrow from other clinics, or patients have to go without their medication in this period. Fern's account:

*Because like now, I can give you an example of our medication – in general, the chronic medication and so on. Now it is the stock take at the depot, and if it is stock take at the depot, orders cannot go through to them. Then the patient has to go without medication or we have to borrow, Baviaans, Midland, Aberdeen, Rietbron [other clinics] or at the hospital because orders cannot now go through to the depot. This is also a problem for us. And that's every time when stock take occurs … and then there is no medication for the patient.* (Fern) [Translated]

Niesie spoke emotionally of how they had only recently learned that part-time MH services by a psychologist was available (filled by the first author since October 2020 at a different secondary care facility in the area) for service users, but that there existed long waiting periods before patients could be seen and treated:

*With uhh psychologists, as previously stated, we only recently found out that uhm, … that the psychologist here at Graaff-Reinet does see patients from outside [implying that the psychologist would see patients from facilities apart from where he is stationed], which we weren't aware of. It was never communicated to us and he is already booked full for a very long period of time [nervous laughter]. So how do you really make use of this multidisciplinary team, with that said, with that being said?* (Niesie)

Anna similarly explained that long waiting times before MH patients can be admitted to specialized institutions are becoming a prominent problem at their facility:

*Psychiatry and mental health issues are having a big snowball, huge effect, because we can't get beds … mental health institutions are full all the time, or are booked, and it sometimes takes longer to stabilize such a patient. So your patient's stay at your referral hospital may be longer, so I, we have half this pile up of people because we are waiting for beds, but remember we are only one facility. How many people rely on that, is it, it's only Elizabeth Donkin who is the state hospital in the Bay [Port Elizabeth – Gqeberha], to which we refer, then there are also sometimes, it's only Elizabeth Donkin who does psychiatry at this stage. It's that whole environment that has to go in there. That's a problem for me. So I don't know of another facility somewhere – now I live in a dream world, there in the Bay where we can refer to, or if you address the problem here, right, then you have less references to the Bay, but … some of these people must really, you understand, they are aggressive, they are, uhm, they are a danger to themselves and society. This type of person, one can't … wait two, three weeks to see a specialist.* (Anna) [Translated]

Communication between health care workers from different facilities or between different levels of care (e.g., primary and secondary sectors) was problematic for many of our participants. Health workers felt that the communication difficulties hampered their knowledge about patients' health status and the continuation of care that they provide:

*I don't know. I'm honest…. And some of them [referring to clinic health care workers] write referrals, but we don't get the back referral in return to say what they have, what happened to that patient [after being treated at hospital level]. And I say, and I think that's why we lose so many, and many of them [patients] that are sent back into the community that is so sick and broken outside.* (Fern) [Translated]

Niesie had similar experiences:

*You will refer a patient and then you won't get feedback from the hospital. They don't. Sometimes they do write, uh refer a letter to you back, and sometimes you just – the patient just vanishes from the face of the earth. So you don't know whether the patient was started on treatment like that. Was the patient then admitted to a tertiary level now? You don't know.* (Niesie)

**Sub-theme 2: Lack of resources.** Health care workers provided numerous instances where diverse and pervasive lack of resources posed challenges to MH care services. Limited infrastructure, clinician person power, restricted scope of psychotropic medication, and few specialist care facilities are some of the difficulties that participants experienced. Diena said:

*No Sir, there is a lack. I think there, I think more can be done for access because in the first place look like now, look if a patient is aggressive, there is not a room – a separate room for example where you can now say that you are now going to take the patient there. Now there is a patient then the patient has to be asked to come out because the patient now has to be treated, examined and to see what the problem is and whatever…. So I don't feel it is, I feel there is still a lot needed for people with a mental deficit.* (Diena) [Translated]

Lily mentioned that there are too many patients for one part-time clinical psychologist to treat at their facility:

*And I think it is … yes. And there are many patients and because our clinical psychologist is not full-time either. So in other words, his abilities are also limited, and he can also only up to a certain, you know, he can also only see a certain number of people in a week.* (Lily) [Translated]

Nadine painted a picture of medication being used as a stopgap for other services:

*Basically, we are only here to distribute medication to keep their conditions stable. We, to actually give them medication that they have already been prescribed. Otherwise it does not exist, because those people are not assessed. There's not even a place here at the clinic for, a place set aside with the facility so that if those people come in, if they maybe, maybe the medication does not work or something, there's not a place where we can … there's not even someone to temp, to attempt that maybe there is a psychiatric nurse who is specialized in the field to care for that person. This is basically why I say and the same happens even at the hospital there are no specialized psychiatric sisters who can look or a doctor, a psychiatrist who can look after such persons and based on this, I say it does not exist.* (Nadine) [Translated]

Lily similarly recounted that there is only one specialist hospital that MH patients can be transferred to from their facility. As a result, patients need to be accommodated for up to three weeks at their hospital before they can be transferred to tertiary level:

*For example, let's think … yes, for example, let's think about patients who are admitted for the seventy-two-hour observation period for certification. Yes, we have one facility, it's Donkin, where these patients go, so in other [words] … I think they also only have a certain capacity. So in other words, if you now … I mean if the person is here now, there is no bed for him, so he is waiting here for three weeks now.* (Lily) [Translated]

**Sub-theme 3: Limited knowledge for dealing with complex cases.** Many health care workers experienced both their own and their colleagues' limited knowledge in the assessment, treatment, or management of more complex mental illness cases:

*I just wanted to say, because it's, remember, a general practitioner, so I mean, his training was general. Where I, let's say for example, my training for psychology and sociology was also more towards the general side, where the social [aspect] was the overall line. Now that's a stupid example. But, yes. So it's quite … and I don't think people realize that people with mental health problems really need specialist care.* (Lily) [Translated]

Niesie was similarly concerned about the lack of MH training for medical doctors:

*I don't know if they are – whether they get, let me say, can I say, after school? Do they get anything? You know, like, in-service training, or whatever with regards to mental health care patients. I don't know. I don't know what's, what, what education they get because when you learn something in school and you need to apply it now, it's a whole different case. Because for me, now you see it in reality, you understand? And I don't know if our doctors that are here with us, if they get the necessary uhm, training. I don't know.* (Niesie)

**Sub-theme 4: Patients who present a physical danger to others.** Many health workers reported encountering aggressive patients, which created potentially dangerous working conditions for staff at their facilities. Anna, for example, recounted:

*So I know of a staff member who resigned a month or three ago, she currently works at a clinic, she's an oldish lady, uhm, I'd say she's in her fifties, almost sixty. Been a very good nurse in the general ward…. And, she told me about two incidents and I physically saw it too, where the one psychiatric patient … I think jumped on her from the bed or something. So her arm was in a sling. She was physically hurt. And … then the other day she came across a patient in the women's ward who wanted to hang herself. So, what there … only, those two incidents that she told me, you understand, made her feel that she didn't want to work here anymore. Because she feels it's not safe for her, remember, she's an oldish lady. Then I also know of a, a male nurse who was physically beaten lights-out by a psychiatric patient. Look, they are very strong and can be aggressive. So why I know so much about the stuff is because I'm often in the ward, I take medicine, I'm on call …* (Anna) [Translated]

Similar examples were given by a number of other participants.

### Theme 4: Means of dealing with and managing mental health care patients

Participants provided examples of ways in which they deal with and manage MH patients within their contexts. This theme, with its sub-themes, illustrates some of the strategies that health care workers utilize when required to service MH patients.

**Sub-theme 1: Pharmacotherapy and relying on sedation.** Lily felt that medication serves a pivotal role in MH treatment, stating that only minimal cases at their facility would be adequately treated if medication was not part of the treatment regimen:

*Oh, and then we refer to the doctor, because yes, I believe that a medication can sometimes be to that person's advantage. In exceptional cases, but it doesn't happen very often, then a person will say, but they're all right. They're fine with, like, therapy sessions and they feel better and they don't feel the need for medication. But in most cases, people will kind of buy into the idea of, you know, getting something for anxiety along with the therapy, or yeah.* (Lily) [Translated]

As a medical officer, Tikos is authorized to prescribe sedatives as part of their treatment of MH patients. In explaining the reasons for this heavy reliance on sedatives, Tikos spoke of an instance in the past in which another health care worker was admitted for psychiatric purposes:

*At the time the patient behaves like that…, you, you, you don't – it's, it's such an embarrassing act, even to the family…. Unprovoked movements, you know, and as a big guy, you see him doing this and making push-ups, you are worried if he decides to harm us, you know. So a sedation becomes important because you, you really want what, what he is doing to stop. Yes, he was not aggressive but he kept on saying things that, don't tell him, don't you know…. So, is important to us really, to, to – and it gives us time to sort of find more history. What – go back. Let's collect history. What has been happening … and also when we have to refer, when we have to refer it gives us that comfort that he is now sedated. He can go without, without disturbing.* (Tikos)

Niesie explained a case where they felt pharmacotherapy may be over prescribed and possibly hinder comprehensive psychiatric evaluation:

*The patient is paranoid now. The patient is scared. The patient is a known schizophrenic. This patient needs to be admitted for 72 hours to be assessed and during the 72-hour period, it doesn't necessarily mean, mean the patient needs to be sedated now, because that is also something we've, we've, we've picked up from the hospital. For that whole 72 hours, I don't know how the assessment is being done. Uhmm, but the patient is then being sedated…. How are you going to pick up anything if the patient is sedated?... because with mental health, you need to monitor that patient for that 72 hours. Whether the patient is aggressive, or whatever. So how are you gonna do that if you sedate the patient the whole time? What are you gonna pick up? The patient is sleeping for three days, I mean… I don't say they shouldn't because some of the patients are very, very, very uhm, dangerous. So, they can be sedated for that period of time and then wait for the sedation to wane off, to wean off and then see how the patient is. So I don't say they shouldn't sedate the patient…. Sedation is like your last resort, I would say.* (Niesie)

Tikos reported that many of the patients they treat using psychotropic medication to cope with stressors they experience:

*I must mention that in this place, if you can look at the scripts, there is a lot of patients on amitriptyline. More than in other communities where I've worked. People look at this patient who had post-traumatic stress neh. She could quickly take tablets and you know. During COVID times, we had fifteen, sixteen year olds were asking for Phenergan and other drugs for sleeping. So the community lives on sleeping tablets and antidepressant and, and, and…* (Tikos)

**Sub-theme 2: Long distance consultation and care delivery.** Health care workers spoke of experiences of clinicians and treatments not always being available to service users at their facilities, and their frequent reliance on 'long-distance' care approaches to provide access to care for patients.

Participant X explained that they do not have a psychiatrist employed at their facility. As a result, the health worker needs to make telephonic contact with a specialist at another care facility who would then advise on treatment without physically seeing the patient. Participant X described the way in which this compromises the quality of treatment provision:

*We also don't have a psychiatrist who is on the premises who can help with this – we always have to call someone but then they don't see the patient. We just have to, they just work on what we tell them or what the doctor tells them. The patient is so and so and so, and then they're going to tell you to do so and so, give the medication or prescribe it or do it. Where they can sit face-to-face with the patient and see the patient themselves, then perhaps the patient will receive better treatment.* (Participant X) [Translated]

Niesie made a similar comment:

*We don't have stationed psychologists here. We don't have stationed social workers. Yeah. So all our sources or resources that we need for mental health is outside of, of, of primary health care. We don't have anyone stationed … and that's why sometimes I, me personally, I jump them because I have a, a, a link with Professor X (the psychiatrist not on site but servicing the area). So my level in communication would be straight to Professor X instead of still refer- ring the patient to, to the hospital.* (Niesie)

Fern told of how some psychiatric patients are required to walk to a different facility to get their treatment when the clinic does not have the medication available:

*Why don't they get a pharmacist here at the clinic? So that it is not necessary for our patients, if perhaps they don't have the medication … to still go up to the hospital for their medication, because we don't keep their medication now. Their medication comes already packed, ready for them here. So why does the patient have to be so tired out to go up here to the hospital? If they have a pharmacist here and we can keep the medication here, why?* (Fern) [Translated]

**Sub-theme 3: Desperate improvising.** Many participants improvised their approach to service delivery in order to deal with the MH care demands in their context. Tikos explained that not having a specialist MH worker to advise them leads health workers to desperately improvise and introduce treatments that they are not completely familiar with or know the possible effects it may have on a particular patient:

*There's no mental health specialist we can refer to* [at their facility]. *No. Zero, as, as we speak…. With a patient coming to you for help, you know, it's, it's, it's really – it makes us to start –I wouldn't say irresponsible because you are in an act of trying to do something for the patient but you, you're not quite sure of the end result…. The absence of a specialist to talk to me, probably makes us to sometimes try concoctions that we're not sure what will they come up with.* (Tikos)

**Theme 5: Helpful strategies and proposed changes for more effective treatment**

Many participants identified strategies that may have the potential of mobilizing changes that can possibly improve the current status of mental health care services at their facility and in the wider community.

**Sub-theme 1: Ongoing in-service training on mental health.** Tikos asserted that ongoing in-service training would facilitate their understanding of MH patients:

*Regular training. Regular training for mental health care patients. Uhm, you'll understand that diabetes and mental health are different in a way … frequent or regular in-service trainings or refresher courses around mental health patients.* (Tikos)

Anna similarly reported the value of ongoing in-service training in MH, which had been part of their work program in the past:

*Then what we also always had on our Fridays were the Elizabeth Donkin presentations, which I think were important. We as pharmacists or doctors could go, we always sent someone, then it was, whether it's ADHD [Attention-Deficit/ Hyperactivity Disorder], schizophrenia, depression. It was then presented by the professor at Elizabeth Donkin … it's like CPD [continuous professional development] events and things. Because let's face it, you have to read and attend courses and that type of thing, but we still have to do CPDs, so those were two good things that were in place.* (Anna) [Translated]

**Sub-theme 2: Psychiatrically trained staff post matching and multidisciplinary team approach.** Tikos explained the potential benefits of having a psychiatrically trained nurse stationed at primary health facilities to help with preventative assessment and management:

*A psychiatric nurse at the level of the clinic can highlight things more often so that we know, you know. He or she will be our eyes because they go there to take treatment. He will be our eyes when, when patients are coming for their regular treatments because sometimes the patients don't say, but you can see what the patient is suffering for.* (Tikos)

Lily explained that the allocation of MH trained workers, that have the skills to work in the field, to an appropriate post would be helpful, with more specialized assessments often needed when providing services to MH patients. While MH workers are placed at their facility, they are not placed in the appropriate position:

*We have a very good psychiatric sister who is now in … you know, who works in OPD [outpatient department] …. You know, and I kind of think if you can have someone like that in a casualty situation, for example. You know, where … he doesn't necessarily have to work in that department, but if such a type of patient comes in, you know, if one could perhaps use such a person to perhaps help you with assessment. Because I think psychiatry is a specialist field … so I just think it would make the whole process, could we say, easier, or a little smoother if one immediately had guys with, maybe more knowledge in that field, you know, you can use in your assessment.* (Lily) [Translated]

Niesie emotionally reported the lack of multidisciplinary services at their facility, and the need for such a group to provide holistic care to patients:

*I don't really have anything to say with regards to that because there is nothing to say. Nothing is here. So what can I say? How can I put it? Personally, I feel that our mental health care patients does not get holistic treatment in the sense of having a multidisciplinary team looking after our mental health care patients, and because, and the reason for that is, we don't have stationed, uhm team members for a complete multidisciplinary team here…. So does our patient get holistic treatment? No, they do not get the holistic treatment that they need as a mental health care patient.*

**Sub-theme 3: Community-based education and outreaches.** Anna explained that the current difficult socio-economic circumstances have an impact on people's mental health and that they may benefit from community-based MH education talks:

*These are difficult times in which we live now, I think education is also important. I just experience that people are very anxious, people are very depressed, people have no hope, suicidal, people don't have work, people don't have money, what are they reaching at?…. So, I feel in the community too, I don't know, can one go there to speak at clinics or do talks or at schools to inform people, but yes.* (Anna) [Translated]

Cottonwool similarly strongly expressed the preventative need for community-based MH talks:

*Going into the community and getting some sort of incentive as well. Have like a boerewors (South African sausage) day or a pancakes and whatever to draw the crowd in…. Draw the people and give them a talk on substances, the importance of education, the importance of identifying emotional abuse… – the term survival has changed in 2022. It's not caveman kills you know, kudu and we eat. Like, it is more psyche now. People are very aware of their mental health and their emotional well-being. So, you know, giving talks on how to be more aware, how to deal with things in a better way instead of breaking your child down when they're catching on nonsense. Rather try this approach. Try that approach…. Community outreach. Nb! Nb! Nb!* (Cottonwool)

## Discussion

To our knowledge, the presented work is the first of its kind to investigate access to and provision of MH services, and has reported findings from such a diverse clinician group from primary and secondary care settings in the DBNLM area. It further contributes knowledge to our work within the larger project, published elsewhere [13,36,42]. Considering its novelty, the presented work aimed at providing a focused overview of health care workers' experiences with regards to rural MH services in LMICs, in South Africa, and in the area investigated.

In line with our findings reported in Rall & Swartz in 2024 [13], the presented work can be viewed as confirmation that health workers are burdened with high volumes of work in this part of the country. The integration of MH services at PHC and secondary hospital level introduces multifaceted challenges to access and provision capacities [36]. It seems that there are too few health care workers for the influx of patients since the integration of routine MH treatment at lower levels of service provision, and the quality of care to the MH user population may be compromised as a result. Signs of such strain can also be traced in the emotional tone of the extended narratives provided by our participants when describing their experiences. This challenges the idea that the integration of care would allow for better access to, and provision of MH care at patients' doorsteps [1]. While the integration of health care services in the physical domain for many chronic conditions has been effective, we cannot but wonder if it just isn't too difficult to integrate MH services into lower level facilities. This is so, especially in contexts like the area investigated where, as shown by our sample, not all health care workers are adequately trained in MH and very limited continued professional development or education has occurred around MH over the past years [42], as envisaged by the NMHPFSP 2013–2020. It further seems that ironically, refresher courses were attended only by higher level clinicians like medical officers and not primary health nurses, who are the health workers who see the majority of patients [42]. MH cases often present multiple layers of complexity, as they incorporate many permutations of bio-psycho-social etiologies. Specialists such as psychiatrists and psychologists receive many years of training, and commonly struggle to narrow down a definite diagnosis, but have to treat the low hanging fruits of a list of differential diagnoses instead.

Health care workers encounter a wide range of psychopathology from young and old, in differing degrees of severity, and with multiple etiologies. Our work strongly resonates with other research studies (see, for example, Alabi [14]) which found that, despite the attempts that have been made, and gains thereof, significant treatment gaps in MH care still exist nationally, and according to our findings, particularly in the DBNLM area. While task-shifting approaches in treating common mental disorders with manualized treatments have shown favorable outcomes elsewhere in the country, with the support of project funding, and special training of non-specialist workers to provide treatments such as cognitive behavior therapy and interpersonal therapy, the reality in the Eastern Cape is somewhat different. With limited resources, such as inadequate treatment spaces, lack of knowledge amongst health care workers, and reduced assistance from specialists for guidance, the experiences of health care workers in the DBNLM region raises a critical question: Is the tasking of dealing with MH cases, particularly more complex patients, by health care workers possible, or even feasible, without compromising the quality of care provision to patients and the well-being of our health care workers? Task-shifting in strained systems may not be as selective

as elsewhere in the country [12,17], and can lead to task-dumping [9]. We suggest that this is exemplified in the case of the DBNLM area, in leaving the generalist workers to deal with and treat a wide spectrum of often complex mental pathology, including patients who can present a physical danger to others if not managed properly, for prolonged periods of time, as our research shows. The relationship between health care workers' limited mental health knowledge and its impact on patient behavior – or perceptions thereof, can be regarded as a complex issue experienced by participants in our study. This dynamic may contribute to a cyclical pattern in which health care workers experience themselves as under-skilled or inadequately equipped, while viewing patients as disproportionally ill or challenging. Booysen and Kagee [15] warn that South African generalist practitioners may easily become overwhelmed by mentally ill patients with complex or challenging presentations. Lack of knowledge may predispose health care workers to MH care user stigma [45]. This is further in line with literature that found that unconfident or ill-prepared health workers may be more inclined to stigmatize patients with mental disorders than better prepared professionals [46,47]. Findings from the current larger project [13] report that our generalist practitioners are physically and mentally negatively affected by the demands of their work. Our work is in line with that of Alabi [14], that shows that Eastern Cape health care workers feel their backs are against the wall – they are desperate to help, but have limited ability to effectively deal with complex cases.

The doctors are often the first health care workers to treat a MH patient after they have been screened by, for example, PHC nurses. Their restricted knowledge may limit their confidence and ability to assess, diagnose, and treat effectively. The one part-time clinical psychologist is the only in-person MH-specific trained clinician that may be relatively readily available to provide assistance through assessment, diagnosis, and treatment. This leaves the doctors with limited options, often having to rely, primarily, on pharmacotherapy to manage their patients. Our findings resonate with earlier work carried out by Petersen et al. [16] elsewhere in the country, and suggest that most psychotropic medications are readily available at facilities, with medications such as sedatives being heavily relied upon at times, and possibly overprescribed. The findings from our study suggest that health care workers recognize that prescription medication can contribute to more effective treatment and symptom relief, but the more informed health care workers are able to acknowledge that pharmacological interventions may at times be contraindicated, as these can diminish comprehensive assessment and treatment capacities. We also know that medication is not always first line treatment for many stressors/distress and mental disorders.

Despite facing intense challenges, at times our participants have found ways to adapt and work around the challenges. Some of these strategies are sufficient to provide, at least, some form of care. For example, long distance care approaches which may create gaps in treatment (e.g., receiving telephonic guidance from a clinician who did not physically assess the patient), and disruption to those patients who are sent to access required services or treatments at different facilities. In contrast to a process of integration, some of the strategies used in desperation may create more fragmented care, leaving health care workers and patients in a sink or swim position, rather than receiving a product from a one-stop shop. Furthermore, providing MH services in this area, at times, demands that health care workers bend to such an extent to achieve a degree of functionality as to create varying degrees of discomfort for both professionals and patients alike. Our findings resonate with our earlier work [13] which suggests that some staff who feel they are under enormous pressure may be discouraged to continue their services in MH care. Some staff, we found, have actually relocated to other facilities or departments to avoid experiencing such discomfort. Fewer health care workers willing to work in MH likely leads to higher care burden to the remaining few, which inevitably diminishes the quality of access to and provision of MH care in the area.

Many of our participants are motivated to improve MH services in the area. The findings provide us with their clear needs which would enable them to facilitate the quality of access to, and provision of MH care in the area. The question remains whether such adaptations are do-able, or if they currently remain locked up in policy and restricted by the systematic challenges of the public health sector.

## Conclusions and recommendations

The decentralization and integration of MH care into lower care sectors makes sense in theory, but the reality experienced by health workers in this area demonstrates a situation of great stress and, often, neglect. Recent findings suggest that up to 45% of the country's MH budget is allocated to specialist psychiatric facilities (i.e., tertiary care), compared to a mere 7.9% to community, PHC, and outpatient services [5]. Lund and Flisher [48] reported that up to 34% of contacts with MH service users occur during inpatient admissions. The question arises as to how the majority of MH patients can be cared for when MH services are de-institutionalized and integrated at primary and secondary level, with an extremely limited budget? The influx of patients and limited care providers introduces challenges to both, health care workers and patients and the appropriate structuring and management of MH patients and services are crucial. It is difficult to formulate concrete local recommendations against a backdrop of various ongoing multifaceted challenges and complexities that are influenced by broader issues, which affect access to and provision of MH care county wide. Interventions need to be considered in the context of broader service provision contexts and issues. Rall & Swartz [36], for example described that basic interventions such as "resegregating care" and scheduling separate appointment slots for MH patients have been found to be helpful in this setting, despite the fact that in principle the integration of MH into PHC would argue against separating services. Participants further suggest that MH services should be expanded to community level (e.g., MH education programs and specialist outreach programs), but their suggestions are unlikely to be implemented in the context of budget restrictions, an issue which must be addressed. Our study provides encouragement for increased budgets to fund resources, such as the appointment of more MH trained staff and assigning them to appropriate service positions. The development of community-based MH services should be considered to assist health care workers and patients.

A previously strong outreach program by specialist psychiatry staff from tertiary level facilities is greatly missed in our study area. Van Heerden et al. [28] have noted the benefits, and some challenges, regarding psychiatric outreach programs. The value and need for specialist support can, however, not be ignored here. Our findings suggest that there needs to be rethinking around restructuring this type of services (e.g., compulsory psychiatric community service and community MH services) in South Africa.

Currently, many patients are discussed telephonically by health care workers consulting with specialists, but these specialists may never see the patients in person. There are no telehealth services where the specialists may directly observe and interact with patients. What treatment gaps are created when self-acknowledged generalists claim that they do not possess sufficient knowledge to deal with complex cases? What are they able to communicate to an off-site specialist, when, and if, they are able to contact them telephonically?

The personnel we interviewed work in a system which notionally has training available in MH, and resources for frontline personnel. The reality presented to us was that these features of the system are not there. It should strongly be considered to reintroduce refresher courses and continuous in-service training in MH again, to all relevant health care staff, but especially at lower levels of care where exposure to MH educations seems especially limited. The issues faced are resource-based and structural, and without the support and training available in reality, patient care is compromised and care providers are under considerable strain. The mismatch between what is theoretically available and what is actually provided and not provided is worrying, and all MH advocates and specialists should be engaged in advocacy about this. Ideologies of inclusion ring hollow when inclusion does not in fact occur.

## Supporting information

**S1 Text. Healthcare provider participant demographic questionnaire.**
(DOCX)

**S2 Text. Healthcare provider qualitative interview schedule.**
(DOCX)

## Acknowledgments

Thank you to the participants for your contribution to our understanding of your experiences of providing mental health care in this part of our country. Thank you Mr Cyril Clarke for your assistance with the translation of the narratives. Thank you Ms Jacqueline Gamble for your valuable inputs in formatting and editing the final draft to be presented.

## Author contributions

**Conceptualization:** Divan Rall.

**Data curation:** Divan Rall.

**Formal analysis:** Divan Rall.

**Investigation:** Divan Rall.

**Methodology:** Divan Rall.

**Resources:** Leslie Swartz.

**Supervision:** Leslie Swartz.

**Writing – original draft:** Divan Rall.

**Writing – review & editing:** Leslie Swartz.

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
