## [Decision Letter · Decision Letter 0]

10 Jul 2025

PMEN-D-25-00185

‘The community lives on sleeping medication and antidepressant[s]….’: Health care workers’ experiences of mental health service provision in rural South Africa

PLOS Mental Health

Dear Dr. Rall,

Thank you for submitting your manuscript to PLOS Mental Health. After careful consideration, we feel that it has merit but does not fully meet PLOS Mental Health’s publication criteria as it currently stands. Therefore, we invite you to submit a revised version of the manuscript that addresses the points raised during the review process.

We look forward to receiving your revised manuscript.

Kind regards,

Sauharda Rai, Ph.D

Academic Editor

PLOS Mental Health

Journal Requirements:

1. Please send a completed 'Competing Interests' statement, including any COIs declared by your co-authors. If you have no competing interests to declare, please state "The authors have declared that no competing interests exist". Otherwise please declare all competing interests beginning with the statement "I have read the journal's policy and the authors of this manuscript have the following competing interests:"

2. We noticed that you used “unpublished data” in the manuscript. We do not allow these references, as the PLOS data access policy requires that all data be either published with the manuscript or made available in a publicly accessible database. Please amend the supplementary material to include the referenced data or remove the references.

3. In the online submission form, you indicated that “All data are stored on the password protected computer and in locked cabinets of the PI. The datasets used and/or analyzed during the current study are available from the corresponding author on reasonable request.”. 

3. Uploaded as supplementary information.

Additional Editor Comments (if provided):

This is a timely and important paper, especially given the growing global momentum to integrate mental health into primary healthcare. The authors effectively highlight the perspectives of frontline health workers—the key drivers of these initiatives—offering valuable insights for policymakers, researchers, and program implementers. Their work provides critical food for thought on how best to operationalize this integration. Besides the comments from the reviewers here are my comments

1. The method section needs to be stronger. Can the authors provide more information on data analysis process. Was the code development/coding process done only by one person i.e. the first author? Who conducted these interviews?

2. Getting more information of the participant would provide more richer perspective to the result section. Besides what you have in table 1, did you collect data on things like if they received any mental health training? If yes, please add

3. My understanding is that the first author was a part of the system, as a psychologist. Can the authors add issues of personal biases, positionality as you conducted the research as well as during analysis and writing phase. How was the author’s own experience being a provider in the study area impact this paper?

4. As a stigma researcher, it was interesting to read about Theme 3- esp. subtheme 3 and 4 which fits well on how we define provider’s stigma. (Kohrt et al 2021 - https://doi.org/10.1016/j.socscimed.2020.112852) The challenges faced by the health workers as well as the environment the authors explains how these forms of stigma is manifested. Can the authors expand more on this?

5. It was also interested to read how health workers are relocating because of the challenges they are facing. This is exactly the opposite of what task sharing is trying to achieve – i.e. reducing treatment gap in the absence of health workers. Can the authors add more of this in the discussion section.

6. Overall, I feel the challenges authors describe here and the critical questions they ask applies to other settings too, esp. low-resource settings where the ultimate burden is falling upon health workers who are already over stretched. So I would suggest the author to consider expanding the scope of this paper – bringing in global perspectives and how the findings here imply to other similar places.

Reviewers' comments:

Reviewer's Responses to Questions

**Comments to the Author**

1. Does this manuscript meet PLOS Mental Health’s publication criteria?

Reviewer #1: Partly

Reviewer #2: Yes

2. Has the statistical analysis been performed appropriately and rigorously?

Reviewer #1: N/A

Reviewer #2: N/A

3. Have the authors made all data underlying the findings in their manuscript fully available (please refer to the Data Availability Statement at the start of the manuscript PDF file)?

Reviewer #1: Yes

Reviewer #2: Yes

4. Is the manuscript presented in an intelligible fashion and written in standard English?

Reviewer #1: Yes

Reviewer #2: Yes

Reviewer #1: This is a valuable and timely qualitative study exploring frontline health care workers’ experiences with mental health (MH) service provision in a rural, under-resourced region of South Africa. The manuscript offers rich narratives that shed light on structural challenges, resource limitations, and adaptations made by generalist providers working in the absence of adequate specialist support. I have a few minor comments:

1. Methodological Transparency: The abstract suggests qualitative methodology but lacks detail. Recommend elaborating on sampling methods, coding processes, reflexivity, and how trustworthiness (e.g., triangulation, audit trail) was ensured.

2. Analytical Rigor: Clarify whether thematic saturation was reached, whether coding was independently verified, and if software (e.g., NVivo) or established procedures were used.

3. Literature Integration: While the study references rural mental health literature, expanding the discussion to include broader qualitative analyses of psychotropic overuse and task-shifting in LMICs would enhance generalizability.

4. Policy and Practice Implications: The manuscript would benefit from more concrete recommendations. For example, how can deprescribing frameworks, psychosocial supports, or community outreach models be practically piloted in similar contexts?

Reviewer #2: Abstract

Health workers are burdened with high volumes of work in this part of the country. -repetitive in the previous sentence

Introduction – would benefit from understanding the statistics or prevalence of mental health in SA. What are the policies related to mental health from the national department of health perspective? What services are supposed to be offered at the secondary and primary health care levels for mental health? This would create a better background for the reader

Clarify if you are referring to acute psychological intervention or holistic services which include rehabilitation and integration. This will help the reader contextualise the student

Methods

How is this submission different from any other submission on the larger project that investigated aspects of access to and provision of public MH services? Given that mixed methods have to integrate data when reporting on the results.

Research setting – needs to be explained as this is a international journal.

Data analysis

Transcribed narrative – does this refer to transcript- this may confuse the reader

Results

Valium- please change the pseudonym as this is a drug used- may confuse the reader

Quote not clear. Person states that they encounter persons who are suicidal, but quote says there wouldn’t be such high incidences of people committing suicide

Niesie reported that they also encounter persons who become suicidal because they are

236 depressed: ‘There wouldn’t be such high incidences of people committing suicide … when

237 they feel very depressed or low ….’

Discussion

Task shifting is a reality in low and middle-income countries at the secondary and tertiary levels. What is the training of the current professionals you interviewed, and what is the gap? This will help the reader understand the context and the gap in services

**Do you want your identity to be public for this peer review?** For information about this choice, including consent withdrawal, please see our Privacy Policy

Reviewer #1: No

Reviewer #2: **Yes: ** Deshini Naidoo

---

## [Editor Report · Decision Letter 1]

23 Oct 2025

‘The community lives on sleeping medication and antidepressant[s]….’: Health care workers’ experiences of mental health service provision in rural South Africa

PMEN-D-25-00185R1

Dear Mr Rall,

We are pleased to inform you that your manuscript '‘The community lives on sleeping medication and antidepressant[s]….’: Health care workers’ experiences of mental health service provision in rural South Africa' has been provisionally accepted for publication in PLOS Mental Health.

Best regards,

Sauharda Rai, Ph.D

Academic Editor

PLOS Mental Health